# Natural Populations of *Astrocaryum aculeatum* Meyer in Amazonia: Genetic Diversity and Conservation

**DOI:** 10.3390/plants11212957

**Published:** 2022-11-02

**Authors:** Santiago Linorio Ferreyra Ramos, Maria Teresa Gomes Lopes, Carlos Meneses, Gabriel Dequigiovanni, Jeferson Luis Vasconcelos de Macêdo, Ricardo Lopes, Alexandre Magno Sebbenn, Rogério Freire da Silva, Therezinha de Jesus Pinto Fraxe, Elizabeth Ann Veasey

**Affiliations:** 1Instituto de Ciências Exatas e Tecnologia, Universidade Federal do Amazonas, Rua Nossa Senhora do Rosário, 3863, Bairro Tiradentes, Itacoatiara 69100-000, AM, Brazil; 2Faculdade de Ciências Agrárias, Universidade Federal do Amazonas, Avenida Rodrigo Otávio Ramos, 3.000, Bairro Coroado, Manaus 69077-000, AM, Brazil; 3Programa de Pós-Graduação em Ciências Agrárias, Departamento de Biologia, Centro de Ciências Biológicas e da Saúde, Universidade Estadual da Paraíba, Rua Baraúnas, 351, Bairro Universitário, Campina Grande 58429-500, PB, Brazil; 4Centro Universitário de Cascavel, Avenida Tito Muffato, 2317, Bairro Santa Cruz, Cascavel 85806-080, PR, Brazil; 5Campo Experimental da Embrapa Amazônia Ocidental, Embrapa Amazônia Ocidental, Km 29, AM 010, CP. 319, Manaus 9010-970, AM, Brazil; 6Seção de Melhoramento e Conservação Genética Florestal, Instituto Florestal de São Paulo, Rua do Horto, 931, Bairro Horto Florestal, São Paulo 01059-970, SP, Brazil; 7Departamento de Genética, Escola Superior de Agricultura Luiz de Queiroz, Universidade de São Paulo, Av. Pádua Dias, 11, Bairro São Dimas, Piracicaba 13418-900, SP, Brazil

**Keywords:** tucumã-do-Amazonas, molecular marker, populations genetics

## Abstract

*Astrocaryum aculeatum*, a palm tree incipiently domesticated from upland ecosystems in the Brazilian Amazon, is especially adapted to anthropized areas. The pulp of the fruit, obtained by extractivism, is consumed fresh by the Amazonian population. The objective of the study is to evaluate the diversity and genetic structure of the natural populations of *A. aculeatum*, exploited by extractive farmers in Amazonas, Brazil, seeking to suggest conservation and management strategies for this species. A total of 218 plants were sampled in 15 populations in 14 municipalities in the state of Amazonas, evaluated by 12 microsatellite loci. A total of 101 alleles were observed. The means of the observed heterozygosities (*H_O_* = 0.6390) were higher than expected (*H_E_* = 0.557), with high levels of heterozygotes in the populations. The fixation index in the loci and populations was negative. The *F_ST_* (0.07) and AMOVA showed moderate population structure. Bayesian analysis indicated the grouping *k* = 4 as the most adequate. There is a high genetic diversity in populations, with a moderate genetic structure due to possible historical events, which could be related to the process of subpopulation formation, possibly presenting three historical moments: before and after the beginning of deforestation and today. The conservation and management policies of this species must be carried out at a watershed level.

## 1. Introduction

The tucumã-do-Amazonas (*Astrocaryum aculeatum* Meyer), a palm tree of the Arecaceae family, exists in the western and central areas of the Brazilian Amazon. It is distributed in the states of Acre, Mato Grosso, Rondônia, Roraima, part of Pará, and Amazonas [1,2]. Production chain has generated employment and income for the population that inhabits the capital Manaus and the locations where this palm is found [2]. The tucumã-do-Amazonas fruit has nutritional and medicinal properties [3,4,5]. However, the main importance is the fresh consumption of the pulp or elaboration of different products, as well as the preparation of sandwiches, much appreciated by both the local population and tourists. Oils can be extracted from the seeds and the pulp itself mainly for use in the cosmetics industry. The endocarp is used in handicrafts [1,2,6]. Almonds can be used as an input for the production of biofuel [7,8] or as a catalyst oil for the electrolytic paste of used batteries [9]. Ecologically, this species can recover degraded areas due to adaptation to upland ecosystems and, more commonly, deforested areas or areas that have suffered some anthropic action [1,10]. It also attracts species of the Amazon fauna, such as *Dasyprocta azarae* and *Myoprocta* sp., which end up behaving as secondary dispersers (zoochoric) of the species itself [11] and leading to natural recovery in some areas by bringing seeds of other species of different ecological succession.

Extractive exploitation of *A. aculeatum*, a **n**on-**t**imber **f**orest **p**roduct (NTFP), is the main available means of production. It is irregularly offered, in both quantity and quality, to the market of Amazonas, the largest state in Brazil. However, if we analyze the extractive exploitation based on the protection of the Amazon forest ecosystem [12], the conservation of tropical biodiversity and the sustainable management of NTFPs, especially in poor regions, are alternatives for social and economic development of local populations [13]. The ecological impacts of NTFP exploitation may take time to be perceived and require alternative measurement approaches [14]. This is because all fruits are used at the time of collection to then form part of the entire regional production chain of *A. aculeatum* [13].

The knowledge of the parameters of interpopulational and intrapopulational diversity and genetic structure of natural populations of *A. aculeatum* are paramount for the establishment of adequate strategies in the use of this genetic resource. This may be the starting point to obtain a response from current natural populations and it may suggest how to increase the efficiency of domestication and selection at multiple stages of the genetic improvement program for this species. Natural populations of *A. aculeatum* have a great genetic variability of traits, such as plant height, fruit yield and quality, size, pulp yield, flavor, fiber, and oil content [10,15]. However, in addition to morpho-agronomic information, the study of population genetics using microsatellites or simple sequence repeats (SSR) allows estimating genetic variation and population structure [16]. SSR markers were developed for *A. aculeatum* by Ramos et al. [17]. They have been used to assess spatial genetic structure, genetic diversity, and pollen dispersion in a harvested population of this species [11] and to assess the populational genetic structure of three species of *Astrocaryum*, including *A. aculeatum*, originating in the state of Pará [18].

Considering that this species has not yet been domesticated and aiming to contribute to the management, use, and conservation in situ and ex situ of the genetic resources of this species, this research aims to evaluate the diversity and genetic structure of natural populations of *A. aculeatum* Meyer exploited by extractive farmers in the state of Amazonas, Brazil, using microsatellite markers. The following questions were posed: (a) what is the level of genetic variability among the sampled genotypes in the evaluated natural populations of tucumã-do-Amazonas? (b) How is this diversity structured? (c) What is the role of hydrographic basins in structuring populations? (d) Is there isolation by distance between populations? The answer to these questions will enable the improvement and efficiency in the management, use, and conservation of this natural resource.

## 2. Results

### 2.1. Genetic Diversity Indexes

Of the 12 microsatellite loci used, the primers Aac01 and Aac13 were excluded because they presented monomorphism in the plants sampled. The ten analyzed loci were in Hardy–Weinberg equilibrium (*EHW*) (*p* > 0.005) (Appendix A), and the analysis of loci pairs showed 7.7% loci pairs with linkage disequilibrium. The loci were all polymorphic, with a total of 101 alleles, ranging from 3 (Aac02) to 21 (Aac09) alleles per locus, with an average of 10.1 alleles per locus. The expected heterozygosity (*H_E_*) varied between 0.096 for locus Aac02 and 0.87 for locus Aac04. In most loci, the observed heterozygosities (*H_O_*) showed higher values than those of *H_E_*, with the exception of the loci Aac06 (0.63), Aac07 (0.65), Aac09 (0.49), and Aac14 (0.61). *H_O_* ranged from 0.10 for locus Aac02 to 0.97 for locus Aac12. The mean of all loci for *H_O_* was 0.64, higher than the mean for *H*_E_ (0.59).

For populations, the mean number of alleles was 4.5, ranging from 3.5 for the Presidente Figueiredo population–Rumo Certo community (PF–Rumo Certo) to 5.1 for the Humaitá population (Table 1). The values were high for the observed and expected heterozygosities in populations, with *H_O_* showing higher values than the *H_E_* in 14 populations. There was a single exception for the Itacoatiara population, which had a similar value (0.55) to *H_O_* and *H_E_*. The *H_O_* ranged from 0.55 for the populations of PF– Rumo Certo and Itacoatiara to 0.74 for Presidente Figueiredo–Estrada Balbina km 42 (PF–Est. Balbina km 42). The *H_E_* ranged between 0.42 for the PF–Rumo Certo and 0.59 for the Iranduba population. The mean of *H_O_* of all populations was 0.64, thus, superior to *H_E_*, with 0.54. The fixation index (*f*) presented values below zero, which shows an excess of heterozygotes (Table 1).

The analyses also showed a total of 17 private alleles observed in 13 populations. These were obtained for the loci Aac02, Aac04, Aac06, Aac07, Aac09, Aac10, and Aac14 and distributed in different populations (Table 1; Appendix A). The analysis of *EHW* evaluated from the ten loci in the 15 populations showed that 37 interactions between locus and population do not adhere to *EHW* and that the loci Aac03 and Aac10 are those with the greatest lack of adherence to *EHW* in populations. However, the loci Aac02, Aac11, and Aac14 showed *EHW* for all populations. Most loci segregate independently because in the linkage equilibrium analysis, they presented percentages of linkage disequilibrium (*LD*) that varied between 2.22% for the populations of S.S. Uatumã and Iranduba and 24.44% for the population of Borba.

### 2.2. Genetic Structure

The results of the analysis of Wright’s [19] F statistics on the 15 populations of A. aculeatum sampled indicated that the total inbreeding (*F_IT_* = –0.0714) and the estimate of inbreeding due to the reproductive or intrapopulation system (*F_IS_* = –0.1521) were lower in relation to inbreeding due to subdivision (*F_ST_* = 0.0700). The fixation indices *F_ST_* were outside the upper and lower limits of bootstrapping, which indicates that the estimates are significantly different from zero (Table 2). However, the *F_IT_* and the *F_IS_* were not significant.

The pairwise analysis of *F_ST_* between populations showed that there is genetic differentiation for most populations and that they differ significantly from each other, compared (95% CI) to the probabilities of each pairwise comparison (Table 3). The population collected in the municipality of PF–Rumo Certo showed the greatest differentiation, compared to the other populations, followed by the population sampled in the municipality of Novo Aripuanã. The other pairwise comparisons of *F_ST_* showed values lower than 0.09. The pairwise comparison *F_ST_* that showed the smallest difference was between the populations sampled in the municipalities of PF–Est. Balbina km 42 and Manaquiri.

When populations were grouped according to their proximity to a watershed (Appendix A) and compared by the analysis of pairwise *F_ST_*, there was a significant genetic difference between the watersheds. The Amazon and Negro watersheds showed a greater differentiation when compared to each other (Table 4). The pairwise comparison of *F_ST_* with less differentiation between the hydrographic basins was the Amazon and Urubu Rivers. In general, the fixation indices *F_ST_* for populations or watersheds, which were outside the upper and lower limits of zero after bootstrapping, indicate that the estimate is significant. These pairwise *F_ST_* results show that there is genetic structuring present at the population and watershed levels, as observed in divergence values [19]. However, the Mantel test showed a low and non-significant positive correlation (r = 0.2294, *p* = 0.092).

The AMOVA showed the existence of genetic structure in the samples evaluated. Most of the genetic variation occurred within populations (93.62%, *p* = 0.001). The remainder of the genetic variation was observed between populations (6.38%, *p* = 0.001), which means that there is a moderate percentage of differentiation between populations (Table 5).

When estimating the number of genetically homogeneous populations (*K*) through the Bayesian analysis performed by the software structure, two possible forms of grouping were observed: *K* = 3 and *K* = 4 (Appendix A). The main difference between these two clusters is that the cluster *K* = 3 includes the population of the municipality of PF– Rumo Certo and the populations of S.S. Uatumã, Maués, Urucará, and PF–Est. Balbina km 42. However, *K* = 4 separates the population of PF–Rumo Certo from the populations mentioned above and places it as a single cluster, leading to the formation of the fourth cluster. The grouping *K* = 4 is formed by group I (Humaitá, Manicoré, and Novo Aripuanã), group II (Borba, Nova Olinda do Norte, Manaquiri, Iranduba, Itacoatiara, Silves, and Manaus Tarumã-Açú), group III (PF–Rumo Certo), and group IV (S.S. Uatumã, Maués, Ucurará, and PF–Est. Balbina km 42) (Figure 1, Appendix A).

The cluster analysis classified the populations into three groups, which somewhat corroborate the Bayesian analysis (Figure 2). The first group, with three sub-groups, brought together the populations of Iranduba, Manaus Tarumã-Açú, Manaquiri and PF–Est. Balbina km 42, cities very close geographically and connected by roads, for the first sub-group. They are located exactly in the formation area of the Amazon River watershed, between the Solimões and Negro Rivers watersheds, which may explain their greater genetic similarity. A second sub-group with geographic proximity and inhabitants that use the means of fluvial transport through the Amazon, Uatuma, Urubu, and Maués-Açú Rivers, whose fluvial port of departure is the municipality of Itacoatiara (third sub-group), are the populations in the municipalities of Nova Olinda do Norte, S.S. Uatumã, Maues, Silves, and Urucara. In this region, river transport allows the transport of plant material for consumption by the inhabitants of these populations, allowing a genetic flow of the species. In addition, Nova Olinda do Norte (Madeira river), Silves (Urubu river), and Maués (Maués-Açu River) are tributaries to the Amazon River watershed, and these populations are very close to these junctions with the Amazon River. During the flood season, this Amazon River watershed penetrates parts of these areas. The populations of Itacoatiara and Urucará are in the hydrographic basin of the Amazon River.

In the second group of the dendrogram, the populations of Novo Aripuanã, Borba, Humaitá, and Manicoré belong to the Madeira River watershed, showing that there is a flow of genetic material between the populations of this watershed. The population of Humaitá presents a greater genetic similarity with the population of Manicoré. The population of Borba grouped with the populations of group II in the Bayesian analysis (Figure 1), and not to group I, which coincides with the populations of this second group. However, the distribution map shows that this population is in an intermediate area between the first and second groups (Figure 1).

The third group classified the population of the Rumo Certo community in the municipality of Presidente Figueiredo, which has a different genetic constitution from the others. This event can be explained by the fact that this population was isolated by the Balbina Dam and because this population is in an area that is a small island within this dam.

## 3. Discussion

This study showed a high mean number of alleles per locus and high levels of genetic diversity for natural populations of *A. aculeatum* in the state of Amazonas. The numbers are comparatively higher than those found in other studies, such as *Astrocaryum murumuru* and *A. paramaca* [18], *Euterpe edulis* [14], including the species being studied here (*A. aculeatum*), for populations in the state of Pará [18]. This high diversity could also be related to the fixation index (*f*), which shows negative values in populations and an excess of heterozygosity, with higher values of *H_O_*, compared to *H_E_* [21]. This information confirms that *A. aculeatum* presents reproduction by allogamy [10] and does not present regenerants by self-fertilization [11]. We ask the question: would this type of reproduction be a strategy for most species in the family? *Astrocaryum mexicanum* [22] or even in other palm species, such as in *Geonoma schottiana* [23], *Phoenix dactylifera* [24], and *Oenocarpus bataua* [25], showed similar reproductive strategies. Regardless of this possible strategy, the upper mean of *H_O_* over *H_E_* of *A. aculeatum* shows a high diversity compared to *Euterpe oleracea* [26]. However, for the populations of Pará, this strategy was not observed for *A. aculeatum* and *A. murumuru*, only for *A. paramaca* [18].

The presence of private alleles in the populations sampled suggests the importance of genetic conservation for this species. A special management in populations with this allelic difference is suggested [27]. An example is always trying to replace regenerants so as not to affect this high diversity. This is because the management of fruit harvest may affect the abundance and distribution of the resource, as well as the growth and regeneration strategy of the species [28]. It is important to highlight that the populations of *A. aculeatum* evaluated are geographically distributed in two types of climate, *Af* and *Am*, according to the Köppen–Geiger climate classification [29].

The behavior of plants in a given habitat always aims at local adaptation, which would be caused by the contribution of conditional neutrality at many independent loci interacting to influence fitness, with alleles from different constraining environments being favored at different loci [30]. This would be a possible answer as to why allele frequencies in the different populations mostly showed adherence to the *EHW* model. However, the *EHW* algorithm does not consider the performance of evolutionary forces, as it is a referential model, except for those imposed by the reproductive process itself [31].

The high genetic diversity observed for *A. aculeatum* within the domestication process is important information, although anthropic extraction may affect the genetic diversity of the species in the future in the process of recruiting new plants in the seedling bank [32]. This is because domestication usually begins with the exploitation of wild plants; proceeds with the cultivation of plants selected in nature, not genetically different from wild plants; and ends with the fixation of morphological and genetic characteristics carried out by human selection [33].

Due to genetic differentiation because of subdivision (*F_ST_*), Wright’s *F* statistics indicated the existence of a moderate genetic structure [31] among the 15 populations of *A. aculeatum* sampled. Most of the genetic variability sampled is found within populations by the inbreeding information observed within subpopulations due to the reproductive system (*F_IS_*). This information is corroborated by AMOVA results, which show that the greatest concentration of diversity occurs within populations. This result may be strongly influenced by the characteristics of the species and by the ability to disperse its genetic material [11], the degree of isolation of the population, the reproductive system [10], and allele diversity [34].

The pairwise *F_ST_* analyses at the watershed level (Table 4) and between populations (Table 3) showed significance for most comparisons, which confirms the presence of structuring genetics. There was a possible isolation between the hydrographic basins of the Madeira and Uatumã Rivers, especially when compared to other watersheds. The Madeira and Uatumã Rivers are geographically opposite and distributed in the southern and northern regions of the Amazon, respectively (Figure 1). This divergence indicates that there may be genetic structuring between populations [19] and that these populations may be grouped according to the hydrographic basins to which they belong. At the population level, if we consider the geographic extension in a straight line between the most distant populations (Humaitá and Urucará), they are 820.87 km away, while the smaller ones (Urucará and S.S. Uatumã) are 25.68 km apart. We could suggest the hypothesis that the flow of genetic material shared between these populations is inversely proportional to the geographic distances between them because individuals are more likely to disperse to nearby locations. It is possible that the geographic distance between populations, or the isolation that watersheds promote between populations through a possible vicariance, could affect the genetic structuring of populations. However, in many species, the amount of gene flow between populations is inversely proportional to the geographic distances between them because individuals are more likely to disperse to nearby locations, an event known as isolation by distance [35], as observed among the populations that are part of the hydrographic basins of the Urubu and Maués Açu rivers in the pairwise analysis of *F_ST_*. The Mantel test shows a positive, but not significant, genetic correlation, suggesting that the allele frequencies obtained in the populations studied do not depend on geographic distances. However, we do consider river transport as one of the main means of transporting people and different products for consumption, including the fruits of *A. aculeatum*, by extractive farmers from the Amazon forests, which could allow a secondary spread of the genetic material to other areas of the Amazon, thus, confirming the result of the Mantel test.

The genetic structure of *A. aculeatum* populations could be related to the vicariance process, as plant populations are often separated from one another by areas of unsuitable habitat over which migration and gene flow are limited [36] or these populations may also be in an evolutionary process independently of each other, considering that the groups of individuals that occupy the different parts of a species’ distribution may evolve relatively independently of each other under the influence of drift and local selection [36]. This suggests that populations of *A. aculeatum* could have started the process of subpopulation formation recently, considering that the species settles in deforested areas [6], and in view of the historical process of deforestation in the Brazilian Amazon that began in the 1960s [37] associated with economic occupation promoted by governmental and political incentives from 1990 onwards [38]. These historical events of deforestation could make it possible to divide the tucumã-do-Amazonas into three historical moments of the development of its subpopulations, namely before deforestation, after the beginning of deforestation, and in the present.

Before deforestation, or in climax-type succession forests, *A. aculeatum* shares its space with a high diversity of plant species and generally results in a low density of all species [39]. In addition to not presenting marked environmental differences, chance may influence which species will germinate and settle in a given area, forming mosaics of species that use the same set of raw materials to support their metabolic functions and are very similar to each other in terms of resource demands, energy sources, method of nutrient uptake, and even biochemistry, in terms of general similarity from one species to another [39]. Thus, before the beginning of deforestation, it would be known as the process of dispersal that could be related to the process of domestication of the tucumã-do-Amazonas, which probably began with the Amerindians [1]. This domestication event of the species may be closely related to the behavior of these Amerindians, especially in the traditional subsistence system, composed of a high diversification of species and building complex agroecosystems, including timber and non-timber products [40]. However, the tucumã-do-Amazonas presents a primary dispersion pattern, consisting of seed rain, generally concentrated in the canopy projection radius of 3.5 m [1,2], and secondary dispersal carried out by accumulating dispersing rodents (agoutis—*Dasyprocta* sp. and *Myoprocta* sp.), which deposit seeds in the vicinity of plants [41], and also by humans, who transport the fruit to consume or sell to other areas through watersheds or routes between communities within the forests [1]. This seed dispersal process is important to determine the colonization of new sites and migration between neighboring populations, especially if it is zoochoric, because the range of seed dispersal can be substantially greater [42].

The second moment could be related to era after the beginning of deforestation in the Amazon in 1960, marked by the process of formation and expansion of large or small natural populations of *A. aculeatum*, specifically in areas where the species was present in the climax forest before being cleared. This formation and expansion in deforested, anthropized areas could be mainly related to the zoochoric dispersion of pyrenes (integument with almond), allowing them to be present in the seed bank of climax forests before being disturbed, since the soils of tropical forests are often considered as the final place where plant diaspores are deposited [43], starting the process of restoration of these anthropized areas. With deforestation through slash and burn, it is normal for seed banks to start the restoration process in these anthropized areas [44]. This process may have led to the hyperproliferation of *A. aculeatum* trees, as is the case of several species of palm trees that are called secondary and invasive species in the Amazon, such as *Astrocaryum acaule*, *Attalea humilis*, *Bactris maraja*, and *Lepidocarium tenue* [45]. Another factor that could have allowed the establishment of the tucumã-do-Amazonas during the formation and expansion process is that, when the practice of cutting and burning is carried out in the deforestation process, the seed bank experiences a great decrease in the density, richness, and viability of seeds [46], reducing competition for resources with other perennial and pioneer species. This loss in the seed bank is due to the effects of burning, where the temperature 7.5 cm above the soil is within a range between 148–593 °C, and at the soil surface this range varies from 67–310 °C, and 1 cm below the ground surface the temperature reaches 48–199 °C [47]. Temperature may cause cracks and fissures in the pyrene integument due to the rapid distension [6] that allows the free entry of water and gases and minimizes possible physical impediments to the development of the embryonic axis [48]. However, *A. aculeatum* is a species capable of withstanding high soil temperatures, resulting from these fires [6].

The third moment is the current situation for populations. It would be marked by the process of secondary dispersion, carried out by humans, leading to the establishment of new populations. In the collections of tucumã-do-Amazonas in different watersheds, the production of agricultural products already domesticated, incipiently domesticated, or collected in an extractive way in the Amazon forests by Amazonian farmers is commercialized among these populations. They use river transport mainly for this purpose, transporting genetic material to other areas of the Amazon. This indicates that this external connectivity variable is a vector of dispersion that affects the movement of seeds, which makes the plant persist, expand, and colonize new habitats [49]. Thus, the populations of *A. aculeatum*, as they are possibly new populations, are still in the process of adaptation and differentiation from each other, since even within continuous populations, environmental heterogeneity may provide an excellent dimension of the genetic structure with the evolution of local adaptation [50].

The possible historical moments and the beginning of the evolutionary process of adaptation could be confirmed by the clusters obtained in the Bayesian analysis. It happened mainly in the only grouping of the population of PF–Rumo Certo. This event could be related to the isolation by vicariance of this population because it is located on an island within the Balbina Dam. The beginning of the construction of this infrastructure was in 1981. The dam was closed on October 1, 1987, leading to the formation of a reservoir that presents a reticulated interconnection arrangement between backwaters, that is, a labyrinth of channels between approximately 1500 islands and 60 tributaries [51]. It allowed a more competitive genotype in this population to sustain a positive growth, standing out from the others [52] due to the environmental heterogeneity that brought a genetic structure with the evolution of local adaptation [50]. This confirms once again that the populations of *A. aculeatum* are relatively new and that they have a vicariant structure.

Regarding the management of the species, this first study on the diversity and genetic structure of tucumã-do-Amazonas indicates that the conservation of the species should be carried out mainly at the level of watersheds, as the results obtained in the different analyses indicated. Studies should also be carried out at the level of conservation in situ/on farm and at the ex situ level in order to start the process of domestication and improvement of plants through different accessions of genotypes in these watersheds. It allows improvements to knowledge and genetic materials for the benefit of traditional farmers in the Amazon and future ventures.

## 4. Materials and Methods

### 4.1. Study Area and Sampling

In the state of Amazonas, most municipalities are close to the banks of different river basins. Thus, river transport in these hydrographic basins is practically the only means of transporting people and different products for consumption, including the fruits of *A. aculeatum* collected by extractive farmers from the Amazon rainforest to supply their products, especially to the market of Manaus, the capital of the state of Amazonas. Fourteen municipalities were selected seeking to fill most of the main hydrographic basins within the state of Amazonas to carry out the respective samplings. The municipalities selected for this study were: Nova Olinda do Norte, Borba, Novo Aripuanã, Manicoré, and Humaitá on the Madeira River; Presidente Figueiredo, S.S. Uatumã and Urucará on the Uatumã River; Iranduba and Manaquiri on the Solimões River; Maués and Silves on the Maués-Açu and Urubu Rivers, respectively; Itacoatiara on the Amazon River; and Manaus on the Negro River (Appendix A). In each municipality, a natural population of *A. aculeatum* was selected. It represented the municipality (Figure 1), with the exception of the municipality of Presidente Figueiredo, where two populations of *A. aculeatum* were selected. In total, 15 natural populations were identified, totaling 218 samples of *A. aculeatum* (Appendix A). These sampling sites are the same as those used by extractive farmers of *A. aculeatum*, who supply this product to the market in the capital of the state of Amazonas.

The plant material collected was a leaflet, which was stored in a previously identified zip lock plastic bag containing silica gel until it could be transported and stored at –20 °C. These collections were carried out following the rules of SisGen (National System for the Management of Genetic Heritage and Associated Traditional Knowledge, Decree No. 8772, of 11 May 2016, which regulates the Law No. 13,123 of 20 May 2015), record n°. A00C6EC. Samples of progenies of the evaluated plants were accessed and currently constitute the ex situ germplasm bank of this species, located at the experimental field of Embrapa Amazonia Ocidental, in Iranduba, Amazonas, Brazil (coordinates: 3°14′56′′ S and 60°13′20′′ W). The access to the genetic heritage of *A. aculeatum* was authorized by the Brazilian Institute for the Environment and Renewable Natural Resources—IBAMA, registration no. 02001.008004/2010-31 (Special Authorization no. 02/2008), under the project titled “Assessment and selection of superior genotypes of peach palm for palm heart (*Bactris gasipaes* Kunth. var. gasipaes Henderson) and generation of technologies for cultivation of tucumã (*A. aculeatum* G. Mey) in the State of Amazonas”.

### 4.2. DNA Extraction and Genotyping

DNA was extracted according to the CTAB 2× (Cationic Hexadecyl Trimethyl Ammonium Bromide) cationic detergent protocol described by Doyle and Doyle [53] and quantified with GelRed dye. The genomic DNA was standardized at a concentration of 10 ng.µL^–1^ to be used for amplification. The 218 samples were amplified by polymerase chain reaction (PCR) using 12 microsatellite loci (Aac01, Aac02, Aac03, Aac04, Aac06, Aac07, Aac09, Aac10, Aac11, Aac12, Aac13, and Aac14) developed for *A. aculeatum* [17]. PCR reactions showed a total volume of 10 μL, containing 10 ng of genomic DNA, 1× buffer (10× standard Taq reaction buffer), 210 µM of each dNTP, 1.5 mM MgCl_2_, 0.16 µM of primer forward and M13 spray (FAM or NED dyes) [54], 0.32 µM of primer reverse, 1.05 U of Taq DNA polymerase (Invitrogen, Carlsbad, California, USA), and 3.49 µL of ultrapure water [11]. These PCR amplifications were carried out in two steps, according to the procedure described in Ramos et al. [11]. PCR products were previously evaluated on 1.5% agarose gel stained with GelRed dye (Biotium, Fremont, CA, USA) in 1× TBE buffer (pH 8.0).

Sample genotyping was performed by capillary electrophoresis on the ABI 3130XL Genetic Analyzer Automatic DNA Analyzer (Applied Biosystems, Foster City, CA, USA). The ET-550 ROX size standard (GE Healthcare, Amersham, Buckinghamshire, UK) was used to determine the size of alleles. Amplified fragments were observed and analyzed using the GENEMAPPER v. 4.0 software (Applied Biosystems, Foster City, CA, USA).

### 4.3. Analysis of Diversity and Genetic Structure

The genetic diversity of each sampled population was obtained using the total number of alleles (*A_T_*), average number of alleles/locus (*A*), observed heterozygosity (*H_O_*), expected heterozygosity (*H_E_*), fixation index (*f*), and Hardy–Weinberg equilibrium (*EHW*). These parameters were calculated using the function divBasic of the package diveRsity [55] on the R platform [56]. Linkage disequilibrium (*LD*), number of private alleles (*Ap*), and the number of null alleles were calculated using the functions *LD*, Nm_private, and the null of the package genepop [57] on the R platform [56], respectively. The *EHW* and *LD* were performed by Fisher’s exact test with 100,000 permutations. The significance level (*p* < 0.05) of *EHW* and *LD* was adjusted with Bonferroni correction [58].

To verify the existence of genetic structure in Wright’s [19] *F* statistics, total inbreeding levels were calculated in individuals from all populations (*F_IT_*), inbreeding index in subpopulations due to the reproductive system (*F_IS_*), and genetic differentiation due to subdivision (*F_ST_*) using the algorithms of Weir and Cockerham [59]. Looking for genetic differentiation between populations, the calculation of two matrices was also performed with the values of *F_ST_* pair by pair. A matrix at the population level and another matrix between the hydrographic basins formed by the respective populations located on the banks of each hydrographic basin or located close to it were built (Appendix A). The calculation of F statistics from Wright [19] and the pairwise *F_ST_* matrices, as well as the significance levels, were evaluated with a 95% confidence interval (95% CI) with 20,000 bootstrappings, using the function diffCalc of the package diveRsity [55].

The pairwise matrices *F_ST_*, in terms of populations and geographic distance, were used to perform the Mantel test [60,61], seeking to determine the correlation coefficient between them. Significance tests were performed with 9999 permutations using the function *mantel.rtest* of the package *ade4* [62,63,64,65]. The geographic distance matrix was calculated using the DIVA-GIS v. 7.5 [66].

The degree of genetic variation according to hierarchical levels between and within populations was analyzed by analysis of molecular variance (AMOVA) [67], as implemented in the GenAlEx v. 6.5 [68,69] using codominant alleles. Significance was assessed by permutation test using 9999 permutations, and significant differentiation between pairs was calculated using the matrix of FST [68,69].

For the genetic structure analysis, a Bayesian analysis was performed to determine the number of clusters within the set of samples evaluated using the software structure [70] configured in the admixture model for its usual application with natural populations. The number of clusters (*K*) was set from 1 to 20, and for each *K*, twenty iterations were performed, with a Burn-in of 100,000 followed by 500,000 iterations Markov Chain Monte Carlo (MCMC). The number of clusters was estimated using the data probability *Ln (ln Pr(X|K))* for the different values of *K* [70]. With the value of *K* selected, a consensus was reached of iterations performed in this cluster through the CLUMPP v. 1.1.2—Cluster Matching and Permutation Program [71]. With the program Distruct v. 1.1 [72], the graphical visualization of the population structure was performed.

A dendrogram was also constructed with the average linkage method between groups (UPGMA = unweighted pair group method using arithmetic averages) at the population level using the Nei’s genetic distances [20]. The confidence level of the clusters was evaluated with 10,000 bootstrappings on the loci of each individual in the populations. To obtain the matrix of Nei’s distances and the dendrogram with bootstrapping, the function aboot of the package poppr, version 2.8.3, was used [73,74].

## 5. Conclusions

The evaluation of *A. aculeatum* populations used by extractive farmers shows that there is high genetic diversity within populations. However, the genetic structure of this species is moderate and occurs, in part, as a function of watersheds. The groupings obtained in the analysis of genetic structure are important for the conservation and management of the species, allowing directing management policies to the watersheds of the Amazon.

## Figures and Tables

**Figure 1 plants-11-02957-f001:**
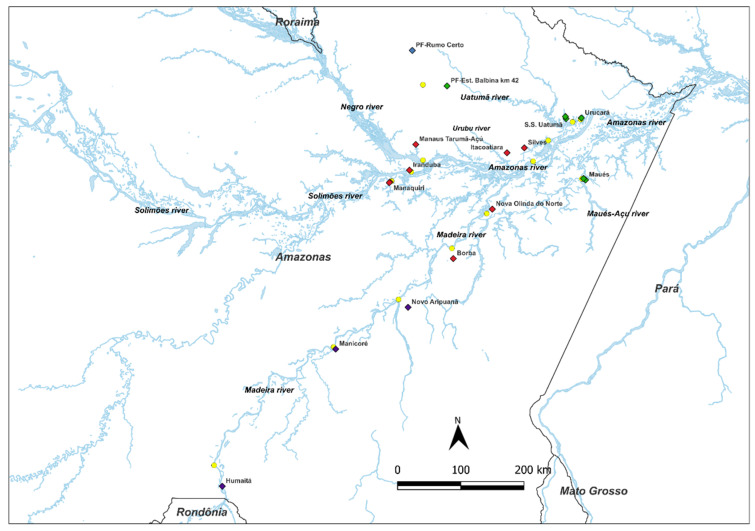
Geographical representation of the genetic composition of 228 plants of *A. aculeatum* used by extractive producers in 15 populations distributed in 14 municipalities in the state of Amazonas, estimated with the *Structure program* from ten microsatellite loci. Each color in the square symbol represents a grouping determined from the estimates *q_K_* obtained from each evaluated plant. Map plotted with DIVA-GIS, version 7.5.

**Figure 2 plants-11-02957-f002:**
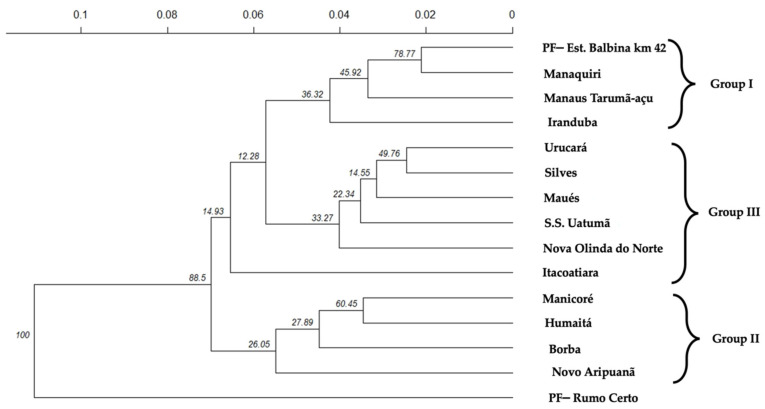
Dendrogram obtained by the “UPGMA” method through Nei’s genetic distances [20] for the 15 populations of *A. aculeatum*. Numbers above the branches of the dendrogram mean the values in clusters are the highest bootstrap results (resampling 10.000).

**Table 1 plants-11-02957-t001:** Genetic diversity indexes at the level of the 15 populations obtained from ten microsatellite loci developed for *A. aculeatum*.

Municipality—Population	*n*	*A_T_*	*A*	*A_p_*	*H_o_*	*H_E_*	*f*
**Humaitá**	15	51	5.1	2	0.64	0.56	–0.131
**Manicoré**	15	48	4.8	1	0.65	0.55	–0.182
**Novo Aripuanã**	15	48	4.8	2	0.63	0.52	–0.203
**Borba**	15	45	4.5	1	0.67	0.56	–0.194
**Nova Olinda do Norte**	15	47	4.7	2	0.68	0.57	–0.204
**Manaquiri**	14	44	4.4	1	0.69	0.52	–0.317
**Iranduba**	13	45	4.5	1	0.67	0.59	–0.143
**Itacoatiara**	15	49	4.9	2	0.55	0.55	–0.015
**Silves**	15	43	4.3	-	0.63	0.53	–0.195
**Maués**	15	44	4.4	1	0.62	0.58	–0.077
**Urucará**	15	42	4.2	1	0.62	0.53	–0.162
**São Sebastião do Uatumã (S.S. Uatumã)**	14	42	4.2	1	0.65	0.53	–0.231
**PF–Rumo Certo**	15	35	3.5	1	0.55	0.42	–0.296
**PF–Est. Balbina km 42**	14	47	4.7	-	0.74	0.56	–0.322
**Manaus Tarumã-Açú**	13	39	3.9	1	0.59	0.50	–0.197
**Average**	-	-	4.5	-	0.64	0.54	-

*n* = number of individuals analyzed by loci; *A_T_* = total number of alleles identified in the population samples; *A* = average number of alleles in the population; *A_p_* = mean number of private alleles; *H_E_* = expected heterozygosity; *H_O_* = observed heterozygosity; *f* = fixation index.

**Table 2 plants-11-02957-t002:** Results of estimates from Wright’s *F* statistics obtained for 15 populations of *A. aculeatum* using ten specific microsatellite loci.

	*F_IS_*	*F_ST_*	*F_IT_*
**Below all loci**	–0.1521	0.0700	–0.0714
**Upper (CI_95%_)**	–0.1277	0.0825	–0.0477
**Lower (CI_95%_)**	–0.1772	0.0587	–0.0950

CI_95%_ = 95% confidence interval.

**Table 3 plants-11-02957-t003:** Pairwise comparisons of *F_ST_* among 15 populations (P) of *A. aculeatum* collected in the state of Amazonas.

Sampled Locations	P1	P2	P3	P4	P5	P6	P7	P8	P9	P10	P11	P12	P13	P14
**P2 = Manicoré**	0.0214													
**P3 = Novo Aripuanã**	0.0557 *	0.0478 *												
**P4 = Borba**	0.0276	0.0427 *	0.0610 *											
**P5 = Nova Olinda do Norte**	0.0557 *	0.0769 *	0.0951 *	0.0442 *										
**P6 = Manaquiri**	0.0460 *	0.0439 *	0.0503 *	0.0317 *	0.0464 *									
**P7 = Iranduba**	0.0705 *	0.0767 *	0.0831 *	0.0586 *	0.0586 *	0.0355 *								
**P8 = Itacoatiara**	0.0718 *	0.0834 *	0.1278 *	0.0512 *	0.0407 *	0.0557 *	0.0629 *							
**P9 = Silves**	0.0711 *	0.0756 *	0.1140 *	0.0374 *	0.0219	0.0285 *	0.0393 *	0.0421 *						
**P10 = Maués**	0.0520 *	0.0735 *	0.1110 *	0.0332 *	0.0342 *	0.0574 *	0.0245	0.0554 *	0.0185					
**P11 = Urucará**	0.0682 *	0.0698 *	0.1041 *	0.0618 *	0.0249	0.0597 *	0.0545 *	0.0633 *	0.0124	0.0177				
**P12 = S.S. Uatumã**	0.0832 *	0.0864 *	0.0883 *	0.0603 *	0.0377 *	0.0656 *	0.0476 *	0.0727 *	0.0335	0.0296	0.0142			
**P13 = PF–Rumo Certo**	0.1535 *	0.1661 *	0.1668 *	0.1807 *	0.1411 *	0.1519 *	0.1594 *	0.1835 *	0.1612 *	0.1556 *	0.1505 *	0.1180 *		
**P14 = PF–Est. Balbina km 42**	0.0517 *	0.0671 *	0.0530 *	0.0561 *	0.0595 *	0.0094	0.0335	0.0845 *	0.0547 *	0.0542 *	0.0734 *	0.0609 *	0.1177 *	
**P15 = Manaus Tarumã-Açú**	0.0729 *	0.0677 *	0.0990 *	0.0585 *	0.0649 *	0.0224 *	0.0248	0.0814 *	0.0520 *	0.0518 *	0.0757 *	0.0683 *	0.1527 *	0.0328

P1 = Humaitá; SS = São Sebastião; * = Significant pairwise comparisons *F_ST_* indicating that there is no difference between populations (with 95% confidence interval with 20,000 bootstrapping.

**Table 4 plants-11-02957-t004:** Pairwise comparisons of *F_ST_* among the seven watersheds where the 15 populations of *A. aculeatum* were collected in the state of Amazonas.

Watershed	RMa	RSo	RAm	RUr	Rmu	RUa
**RSo = Solimões River**	0.0264					
**Ram = Amazon River**	0.0337	0.0331				
**RUr = Urubu River**	0.0402	0.0238	0.0087 *			
**Rmu = Maues-Açu River**	0.0412	0.0339	0.0208 *	0.0185 *		
**RUa = Uatumã River**	0.0439	0.0378	0.0499	0.0508	0.0526	
**RNe = Negro River**	0.0465	0.0127 *	0.0588	0.0520	0.0518	0.0491

RMa = Madeira River; * = Non-significant pairwise comparisons *F_ST_* indicating that there is no difference between watersheds (with 95% confidence interval with 20,000 bootstrapping); watersheds are grouped by populations: RMa (Humaitá, Manicoré, Novo Aripuanã, Borba and Nova Olinda do Norte), RSo (Manaquiri and Iranduba), RAm (Itacoatiara and Urucará), RUr (Silves), RMu (Maués), RUa (Presidente Figueiredo and S.S. Uatumã), and RNe (Manaus).

**Table 5 plants-11-02957-t005:** Analysis of molecular variance (AMOVA) performed for 15 populations of *A. aculeatum* collected in the state of Amazonas.

Source of Variation	df	MS	Est. Var.	PV (%)
**Among populations (Among Pops)**	14	8.4043	0.192	6.38
**Within populations (Within Pops)**	421	2.8200	2.820	93.62
**Total**	435		3.012	100

*p*-value = 0.001 (Estimated based on 9999 permutations). Df = Degrees of Freedom; MS = Mean Squares. Est. Var. = Variance component. PV = Percentage of Variation.

## Data Availability

Not applicable. Institutional Review Board Statement Not applicable.

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
