# Peer review of "Natural Populations of Astrocaryum aculeatum Meyer in Amazonia: Genetic Diversity and Conservation"

_plants, 2022, doi:10.3390/plants11212957_

Round 1
Reviewer 1 Report
The manuscript titled with "Natural populations of Astrocaryum aculeatum Meyer in the 2 Amazonia: genetic diversity and conservation" I see the idea of the manuscript is good and very simple for utilization. The manuscript is written very well and all the materials and method are good and well satisfied for the work. The obtained results are excellent and the analysis is quite enough for the obtained data. Discussion is too long and should be summarized and the same for the conclusion.
Author Response
Dear Review 1,
The constructive criticisms of referees certainly contributed to the improvement of the manuscript. We attended most of the points raised by the reviewers, modified others and addressed detailed answers to those that we do not agree with them. Therefore, its quality has improved substantially and should be at a standard level of the Plants (MDPI). We removed repeated information, explanatory phrases and less important information. We also performed a good synthesis in the conclusion.
The manuscript titled with "Natural populations of Astrocaryum aculeatum Meyer in the 2 Amazonia: genetic diversity and conservation" I see the idea of the manuscript is good and very simple for utilization. The manuscript is written very well and all the materials and method are good and well satisfied for the work. The obtained results are excellent and the analysis is quite enough for the obtained data. Discussion is too long and should be summarized and the same for the conclusion. => We appreciate the compliments for the quality of the work and the interest in our results. Suggestion answered and clarified sentence in the manuscript.
All points were marked in the text below.

Reviewer 2 Report
The study "Natural populations of Astrocaryum aculeatum Meyer in the Amazonia: genetic diversity and conservation" is like a journey on the Amazonas river that fascinates you and leads through exotic nature and culture. The study is well-elaborated, the results statistically processed, and the obtained tendencies are discussed and explained in the biological and cultural context.
Nevertheless, there are minor points that should be addressed
1) Figure 1 (map)
1. Rivers Urubu and Uatuma should be shown in Figure 1.
2. "P." should be removed from the names of populations in the map - this is an unnecessary noise (e.g. "Manicoré" instead of "P.Manicoré")
3. Thre resolution of the map should be increased, in its present form it pixelates.
4. Red and green squears should not overlap the names of populations (e.g. S. S. do Uatumã and Itacoatiara)
5. Pop-up window "Mapa Descricao gerada automaticamente" should be removed
2) The names of populations should be uniform throughout the text, tables, and figures (including map in Figure 1). The names are rather difficult and complex, and it is not easy to follow the story when they population names are shown in different ways. For example, in different places you use:
Manaus (Tarumã-Açú)/Manaus-Tarumã-Açú/Manaus Tarumã-Açú
Presidente Figueiredo (Estrada Balbina, Km 42)/Presidente Figueiredo - estrada Balbina km 42/PF–Est. Balbina km 42/Presidente Figueiredo (Estrada a Balbina km 42)
Presidente Figueiredo (Rumo Certo community)/Presidente Figueiredo - Rumo Certo
I recommend that you use abbreviations for populations titles. For example, "PF–Est. Balbina km 42" instead of full "Presidente Figueiredo - estrada Balbina km 42". Please, once you have chosen the way to designated populations, use it constantly and uniformly.
3) Minor corrections in the text:
line 29 "The objective is to evaluate the diversity" should be "The objective of the study is to evaluate the diversity"
line 92 - not SRS, but SSR (simple sequence repeats) abbreviation is conventionally used.
line 118 "HAND" - did you mean "HE"?
Table 1 Please, use uniformly "HE" or "He"
lines 124-127 the sentence should re-organized in approximately the next way: "The HO ranged from 0.55 for the populations of Presidente Figueiredo (Rumo Certo community) and Itacoatiara to 0.74 for Presidente Figueiredo (Es-125 trada Balbina km 42)". In its present form it is confusing.
line 119-120 "ranging from ... to ..." instead of "ranging from ... and ..."
line 96 "Astrocaryum" should be in latin
4) Figure 2
Please, show groups I (with subgroups), II, and III in the right side of dendrogram. The pop-up window should be removed.
Kind regards,
Reviewer.
Author Response
Dear Review 2,
The constructive criticisms of referees certainly contributed to the improvement of the manuscript. We attended most of the points raised by the reviewers, modified others and addressed detailed answers to those that we do not agree with them. Therefore, its quality has improved substantially and should be at a standard level of the Plants (MDPI).
The study "Natural populations of Astrocaryum aculeatum Meyer in the Amazonia: genetic diversity and conservation" is like a journey on the Amazonas river that fascinates you and leads through exotic nature and culture. The study is well-elaborated, the results statistically processed, and the obtained tendencies are discussed and explained in the biological and cultural context. => We appreciate the compliments for the quality of the work and the interest in our results.
Nevertheless, there are minor points that should be addressed
1) Figure 1 (map)
- Rivers Urubu and Uatuma should be shown in Figure 1. => Suggestion answered and clarified sentence in the manuscript.
- "P." should be removed from the names of populations in the map - this is an unnecessary noise (e.g. "Manicoré" instead of "P.Manicoré") => Suggestion answered and clarified sentence in the manuscript.
- The resolution of the map should be increased, in its present form it pixelates. => Suggestion answered and clarified sentence in the manuscript.
- Red and green squears should not overlap the names of populations (e.g. S. S. do Uatumã and Itacoatiara) => Suggestion answered and clarified sentence in the manuscript.
- Pop-up window "Mapa Descricao gerada automaticamente" should be removed => Suggestion answered and clarified sentence in the manuscript.
2) The names of populations should be uniform throughout the text, tables, and figures (including map in Figure 1). The names are rather difficult and complex, and it is not easy to follow the story when they population names are shown in different ways. For example, in different places you use:
- Manaus (Tarumã-Açú)/Manaus-Tarumã-Açú/Manaus Tarumã-Açú
- Presidente Figueiredo (Estrada Balbina, Km 42)/Presidente Figueiredo - estrada Balbina km 42/PF–Est. Balbina km 42/Presidente Figueiredo (Estrada a Balbina km 42)
- Presidente Figueiredo (Rumo Certo community)/Presidente Figueiredo - Rumo Certo
I recommend that you use abbreviations for populations titles. For example, "PF–Est. Balbina km 42" instead of full "Presidente Figueiredo - estrada Balbina km 42". Please, once you have chosen the way to designated populations, use it constantly and uniformly. => Suggestion answered and clarified sentence in the manuscript.
3) Minor corrections in the text:
line 29 "The objective is to evaluate the diversity" should be "The objective of the study is to evaluate the diversity" => Suggestion answered and clarified sentence in the manuscript.
line 92 - not SRS, but SSR (simple sequence repeats) abbreviation is conventionally used. => Suggestion answered and clarified sentence in the manuscript.
line 118 "HAND" - did you mean "HE"? => Suggestion answered and clarified sentence in the manuscript.
Table 1 Please, use uniformly "HE" or "He" => Suggestion answered and clarified sentence in the manuscript.
lines 124-127 the sentence should re-organized in approximately the next way: "The HO ranged from 0.55 for the populations of Presidente Figueiredo (Rumo Certo community) and Itacoatiara to 0.74 for Presidente Figueiredo (Es-125 trada Balbina km 42)". In its present form it is confusing. => Suggestion answered and clarified sentence in the manuscript.
line 119-120 "ranging from ... to ..." instead of "ranging from ... and ..." => Suggestion answered and clarified sentence in the manuscript.
line 96 "Astrocaryum" should be in latin. => Suggestion answered and clarified sentence in the manuscript.
4) Figure 2
Please, show groups I (with subgroups), II, and III in the right side of dendrogram. The pop-up window should be removed. => Suggestion answered and clarified sentence in the manuscript.
All points were marked in the text below.

Reviewer 3 Report
Analysis of genetic variability and structure is important for the management and conservation of economically and ecologically important species, such as Astrocaryum aculeatum. In my opinion, the study is interesting and relevant, well-executed, and the manuscript can be published.
Some comments:
The research being conducted, both in its methods and aims, is from the field of population and ecological genetics. However, the introduction is written as an ethnobotanical article. I would like this part of the manuscript to reflect more on the problematics of molecular research on tropical tree species.
I suggest delete lines 66-71.
Why is the acronym SRS (simple repeated sequences) used when referring to microsatellite markers when the commonly accepted acronym is SSR (simple sequence repeats). This acronym (SSR) was also used in the article by Ramos et al. 2012, where microsatellite loci were developed for tucumã of Amazonas (A. aculeatum).
Line 118 - HAND ?
Line 241 (Figure 1) change to (Figure S1). However, I would strongly recommend moving Figure S1 to the main body of the article. The information provided in this figure is interesting and should be readily available. Groups I-IV should also be indicated in this figure.
Figure 2. Please indicate in the legend of this figure what the numbers above the branches of the dendrogram mean. Clusters (I-III) should be indicated in the figure for convenience.
Author Response
Dear Review 3,
The constructive criticisms of referees certainly contributed to the improvement of the manuscript. We attended most of the points raised by the reviewers, modified others and addressed detailed answers to those that we do not agree with them. Therefore, its quality has improved substantially and should be at a standard level of the Plants (MDPI).
Analysis of genetic variability and structure is important for the management and conservation of economically and ecologically important species, such as Astrocaryum aculeatum. In my opinion, the study is interesting and relevant, well-executed, and the manuscript can be published. => We appreciate the compliments for the quality of the work and the interest in our results. Suggestion answered and clarified sentence in the manuscript.
Some comments:
The research being conducted, both in its methods and aims, is from the field of population and ecological genetics. However, the introduction is written as an ethnobotanical article. I would like this part of the manuscript to reflect more on the problematics of molecular research on tropical tree species. => Suggestion answered and clarified sentence in the manuscript.
I suggest delete lines 66-71. => Suggestion answered and clarified sentence in the manuscript.
Why is the acronym SRS (simple repeated sequences) used when referring to microsatellite markers when the commonly accepted acronym is SSR (simple sequence repeats). This acronym (SSR) was also used in the article by Ramos et al. 2012, where microsatellite loci were developed for tucumã of Amazonas (A. aculeatum). => Suggestion answered and clarified sentence in the manuscript.
Line 118 - HAND? => Suggestion answered and clarified sentence in the manuscript.
Line 241 (Figure 1) change to (Figure S1). However, I would strongly recommend moving Figure S1 to the main body of the article. The information provided in this figure is interesting and should be readily available. Groups I-IV should also be indicated in this figure. => We appreciate the suggestion. We keep it as presented in the original version and approved by reviewers 1 and 2. We believe that being a supplementary figure will not lose its importance.
Figure 2. Please indicate in the legend of this figure what the numbers above the branches of the dendrogram mean. Clusters (I-III) should be indicated in the figure for convenience. => Suggestion answered and clarified sentence in the manuscript.
All points were marked in the text below.
